# High PD-L1 Expression Predicts for Worse Outcome of Leukemia Patients with Concomitant *NPM1* and *FLT3* Mutations

**DOI:** 10.3390/ijms20112823

**Published:** 2019-06-10

**Authors:** Barbora Brodská, Petra Otevřelová, Cyril Šálek, Ota Fuchs, Zdenka Gašová, Kateřina Kuželová

**Affiliations:** Institute of Hematology and Blood Transfusion, Prague 128 20, Czech Republic; brodska@uhkt.cz (B.B.); Petra.Otevrelova@uhkt.cz (P.O.); Cyril.Salek@uhkt.cz (C.Š.); Ota.Fuchs@uhkt.cz (O.F.); Zdenka.Gasova@uhkt.cz (Z.G.)

**Keywords:** leukemia, AML, FLT3-ITD, NPM1, PD-L1 transcript, PD-1, CD34

## Abstract

Compared to solid tumors, the role of PD-L1 in hematological malignancies is less explored, and the knowledge in this area is mostly limited to lymphomas. However, several studies indicated that PD-L1 is also overexpressed in myeloid malignancies. Successful treatment of the acute myeloid leukemia (AML) is likely associated with elimination of the residual disease by the immune system, and possible involvement of PD-L1 in this process remains to be elucidated. We analyzed PD-L1 expression on AML primary cells by flow cytometry and, in parallel, transcript levels were determined for the transcription variants v1 and v2. The ratio of v1/v2 cDNA correlated with the surface protein amount, and high v1/v2 levels were associated with worse overall survival (*p* = 0.0045). The prognostic impact of PD-L1 was limited to AML with mutated nucleophosmin and concomitant internal tandem duplications in the *FLT3* gene (*p* less than 0.0001 for this particular AML subgroup).

## 1. Introduction

The role of PD-L1 in hematological malignancies is less explored than in solid tumors, and most knowledge is related to the classical Hodgkin’s lymphoma (cHL). In cHL, alterations of the 9p24.1 chromosome locus, which contains the genes coding for PD-L1 and PD-L2, often occur. Moreover, strong overexpression of PD-L1 on the surface of Hodgkin’s lymphoma-specific Reed–Sternberg cells has been documented [1,2]. Concurrently, frequent PD-1 overexpression on tumor-infiltrating lymphocytes (TILs) enhances the likelihood of successful treatment with checkpoint inhibitors. Nivolumab, a PD-1 antibody, has recently been approved for cHL treatment, and additional clinical trials testing the safety and efficacy of other immune checkpoint inhibitors (avelumab, ipilimumab, and pembrolizumab) in cHL are ongoing [3]. Several other lymphoma types, in particular the diffuse large B-cell lymphoma (DLBCL) or the follicular lymphoma, also display enhanced PD-L1 expression correlating with the overall survival or with response to immunotherapy [4,5]. Higher expression of PD-L1 was detected on plasma cells of patients with the multiple myeloma (MM), compared to plasma cells isolated from patients with the monoclonal gammopathy of undetermined significance (MGUS) or to normal plasma cells [6]. The bone marrow microenvironment was shown to induce PD-L1 expression on MM tumor cells [7], and combinatorial checkpoint inhibition, but not the monotherapy, had a therapeutic effect, although it was accompanied with severe toxicities [8]. Several studies reported enhanced PD-L1 expression mediating T-cell exhaustion also in the chronic lymphocytic leukemia [9,10]. In the myelodysplastic syndrome (MDS), the chronic myelomonocytic leukemia (CMML), and the acute myeloid leukemia (AML), increased PD-L1 levels were observed on CD34+ cells, whereas stroma/non-blast cellular compartment was positive for PD-1 [11]. Importantly, in patients treated with an epigenetic therapy, PD-L1 expression, as well as that of PD-L2, PD-1, and CTLA-4, was upregulated, and the expression changes were more pronounced in patients resistant to therapy [11]. Hypomethylating agents (HMAs), such as azacytidine and decitabine, are widely used in myeloproliferative disorders, including leukemias, to activate the transcription of tumor suppressor genes inhibited by an aberrant methylation at their promoters. However, concurrent induction of the interferon response leads to upregulation of immune checkpoint molecules and to secondary resistance to HMAs [12]. A clinical trial with azacytidine and nivolumab combination therapy showed clinical improvement in 52% of HMA naive AML patients, in comparison with 19–25% of patients receiving azacytidine only [13]. Several other clinical trials combining HMAs with PD-L1 inhibitors are ongoing (see e.g., https://clinicaltrials.gov/).

AML is a malignant disease characterized by accumulation of immature hematopoietic cells (blasts) of the myeloid lineage in the bone marrow. An increased amount of immature myeloid elements is usually also present in the peripheral blood (leukocytosis). AML is very heterogeneous as to the causes and prognosis. The main prognostic factors include the age at diagnosis, abnormal karyotype, and a number of molecular markers. The most frequent molecular aberrations in adult AML patients are mutations in the genes for nucleophosmin (*NPM1*), for Fms-like tyrosine kinase 3 (*FLT3*), and for the methyltransferase DNMT3A. Depending on the risk group, the prognosis varies from relatively favorable (about two-thirds of patients surviving three years from the diagnosis) to unfavorable (the overall three-year survival as low as 10%). The standard first-line treatment of AML is based on chemotherapy, which usually induces complete or partial remission. However, the disease symptoms often reappear due to resistant malignant cells (residual disease). It is likely that the immune system plays a crucial role in eventual disease elimination. This idea is supported by the fact that allogenous hematopoietic stem cell transplantation increases the probability of complete cure through graft-versus-leukemia effect of donor immune cells. However, only a part of patients is eligible for this invasive treatment with a high risk of lethal side effects. The current research in AML mainly focuses on novel specific protein inhibitors and on immunotherapy [14]. Possible role of PD-L1 in AML is largely unexplored, although a few reports indicated increased PD-L1 expression on leukemia blasts [11,15,16]. As it was shown in murine models as well as in AML patients, the use of PD-L1 inhibitors should be carefully considered after hematopoietic stem cell transplantation. In this condition, blockade of PD-L1 resulted in increased mortality related to the graft-versus-host disease, likely due to alternative interaction of PD-L1 with CD80 on T-cells [17,18]. 

In general, the impact of high PD-L1 expression on cancer evolution, as well as the consequences of PD-L1 inhibition, remain ambiguous. The prognostic value of PD-L1 levels undoubtedly depends on the tumor type and on other specific conditions. In addition, the method used for PD-L1 expression assessment is important. In the immunohistochemistry (IHC) staining, it is difficult to unify the rating process. Experienced investigators are needed for objective sample evaluation, and the positivity cutoff varies from 1 to 10% among the majority of published studies [19]. In a few cases, the high PD-L1 level was even defined as at least 50% of positive cells (e.g., in [20]). Measurement of transcript levels fits better the requirements for a standardized method. However, four splicing variants are produced from PD-L1 gene (v1 to v4, according to the nomenclature of the National Center for Biotechnology Information, NCBI), three of which coding for proteins, and the function of PD-L1 protein isoforms is not known. We have previously reported that the amount of PD-L1 on the surface of leukemia cells correlated with the ratio of transcript variants v1/v2. In the present work, we confirmed the correlation in 36 AML primary samples. Subsequently, we used the latter parameter for retrospective evaluation of PD-L1 prognostic potential in a larger AML patient cohort (*N* = 86). We found that a high PD-L1 expression on leukemia blasts predicts for worse overall survival (OS), specifically in patients with internal tandem duplications in FLT3 (FLT3-ITD) and with mutated NPM1. 

## 2. Results

Table 1 gives a review of published papers reporting PD-L1 expression on the nucleic acid level (from mRNA or cDNA samples) and usually also on the protein level, the latter being predominantly obtained by immunohistochemistry (IHC). We have gathered information regarding the patient cohort size, the type of the material used, and the disease. When the primer sequences were available, we determined their predicted specificity for different PD-L1 transcript variants using the PrimerBlast software, focusing on the ability to discriminate between the transcript variants v1 and v2. We have included available information about the correlation between PD-L1 transcript and protein levels. Finally, association of PD-L1 expression with the overall survival or with the disease status is given in Table 1, when it was reported. Altogether, the results are rather conflicting with regard to the correlation between the transcript and protein PD-L1 levels, as well as to the prognostic impact of either parameter. The predictive value of PD-L1 expression is likely disease-specific. In addition, the results of the protein analyses might depend on the antibody used [21,22] and, in the case of IHC, on the pathologist’s experience.

We have reported previously that PD-L1 expression on the surface of leukemia cells correlated with the ratio of mRNA transcript variants 1 and 2 (v1/v2) [37]. In the present study, we analyzed the transcript levels from cDNA samples in parallel with the surface protein amount in a cohort of 36 patients with the acute myeloid leukemia (AML) at diagnosis. Samples of peripheral blood leukocytes were obtained from leukapheresis and contained predominantly leukemia blasts, as we determined from CD45/SSC dotplots. The gating strategy is illustrated in Figure 1.

The percentage of blasts in the samples obtained from leukapheretic products was at least 52% (mean: 90%, median: 92%). The signal from PD-L1 antibody usually had rather large intensity distribution on leukemia blasts from the same patient and we thus used both the percentage of positive cells and the mean fluorescence intensity (MFI) for the expression level quantification. PD-L1 expression on lymphocytes was more homogeneous, the positive cell fraction was usually 10 to 20% and MFI ranged from 50 to 100 units. Simultaneously with PD-L1, several other markers related to AML were analyzed on leukemia blasts, and PD-1 expression was measured on T-cells gated as the CD4 or CD8-positive subset of cells in the lymphocyte gate of CD45/SSC dotplots. The transcript level of PD-L1 variants was determined from unseparated samples, which contained both blasts and lymphocytes. However, the contribution of lymphocytes was probably only minor, due to their low amount (about 10%) and smaller size, compared to the blasts. Figure 2 shows plots of PD-L1-positive blast fraction (left column) or MFI from gated blast cells (right column) versus cDNA variant 1, variant 2, or the ratio of both these variants. The results confirmed that v1/v2 transcript ratio from cDNA samples has the best potential to discriminate specimens with high surface positivity determined by flow cytometry in primary AML cells.

To enlarge the patient cohort for evaluation of possible prognostic significance of PD-L1 cDNA, we retrospectively analyzed additional 50 cryopreserved samples from AML patient peripheral blood leukocytes. The set of samples was limited to patients with high blast counts at diagnosis and the samples thus contained mainly leukemic cells. The overall survival data along with PD-L1 v1/v2 cDNA values were subjected to the online Cutoff Finder tool [38], which yielded the optimal cutoff at v1/v2 = 22.72. The survival curves for groups with lower and higher PD-L1 expression are shown in Figure 3a. In the whole cohort, high PD-L1 values were significantly associated with worse survival (*p* = 0.0045, Log-rank Mantel–Cox *t*-test).

As mutations in the gene for Fms-like tyrosine kinase 3 (FLT3) are known to be important prognostic parameters in AML, we further analyzed PD-L1 impact in patient subcohorts divided according to *FLT3* mutational status: 44 patients had wild-type *FLT3* (subcohort denoted as FLT3-WT) and 39 patients had internal tandem duplications (subcohort FLT3-ITD), which are associated with unfavorable prognosis. Three patients (3.5%) had mutations in the tyrosine kinase domain of FLT3, which have unclear prognostic effect, and were excluded from subsequent analyses. The cutoff values obtained using the Cutoff Finder tool for FLT3-WT and FLT3-ITD subcohorts are given in Table 2. The survival curves for patients subdivided according to these cutoff values are shown in Figure 3 (b and c, respectively). The negative impact of high PD-L1 expression was clearly limited to the FLT3-ITD subcohort. Also, a high PD-L1 expression was found in 23% of FLT3-WT patients and in 41% of FLT3-ITD patients (*p* = 0.073 from two-tailed *t*-test, performed using 2 × 2 contingency tables).

Mutations in the nucleophosmin gene (*NPM1*) also belong to important prognostic factors in AML and might be associated with anti-leukemia immune response [39]. We thus also analyzed the impact of PD-L1 expression in subcohorts with different combinations of *FLT3* and *NPM1* mutations. The cutoff values for these subcohorts are again given in Table 2, with the exception of the subcohort NPMwt FLT3-ITD, which did not meet the requirement of at least 20 subjects for analysis by the Cutoff Finder tool. The corresponding survival curves are given in Figure 4, the cutoff being arbitrarily fixed to 22.72 (the value from the whole cohort) for the small NPMwt FLT3-ITD subcohort. Although the patient numbers in the four subcohorts are relatively small, Figure 4 indicates that PD-L1 expression has a very important effect for patients with concomitant FLT3-ITD and *NPM1* mutation. On the other hand, no effect was found in single mutated or non-mutated cases.

As additional data were available for freshly isolated samples from leukapheresis (*N* = 36), we also compared PD-L1 expression on leukemia blasts with PD-1 expression on blasts or on T-cells from the same patient (Figure 5). No obvious correlation was found among these different parameters, neither when comparing the positive cell fractions, nor for MFI values.

Furthermore, it has been reported previously in a cohort of 30 patients that PD-L1 was more frequently expressed in NPMmut cases, in particular in the CD34+ cell compartment. Although our v1/v2 cDNA data also displayed a trend to higher PD-L1 positivity in the NPMmut group (34%) versus NPMwt group (26%), the difference was not statistically significant (*p* = 0.398 from two-tailed *t*-test, performed using 2 × 2 contingency tables). Blasts of NPMmut AML patients are known to be rather CD34-negative. In our patient cohort, the median for the CD34+ cell fraction was 18% for NPMmut patients and 79% for NPMwt patients. However, we did not observe any clear association between CD34 blast positivity and PD-L1 expression (Figure 6).

## 3. Discussion

In general, multiple processes participate in the regulation of protein levels in a cell, and the amount of a protein only partially correlates with the amount of the corresponding transcript [40]. Although assessment of the gene expression at the protein level is more informative and allows for more straightforward interpretation, quantitative measurement of proteins, which is almost exclusively based on antibodies, tends to be less robust and technically more demanding. Therefore, methods based on transcript quantification are often preferred, especially for routine screening. PD-L1 is a transmembrane protein, which undergoes multiple posttranslational modifications, such as N-glycosylation at the extracellular part [41] or palmitoylation at the cytosolic part [42]. Its stability is further affected by interaction with other membrane proteins [43]. In addition, the primers used for the transcript level quantification often do not distinguish among different PD-L1 transcript variants (Table 1). Thus, it is not surprising that no unequivocal conclusions could be drawn so far as to the prognostic value of PD-L1 transcript levels in different tumors.

In hematopoietic cells, protein expression levels on the cell surface are usually determined by flow cytometry, which provides more objective quantification compared to immunohistochemistry. We have previously reported, that PD-L1 expression on the surface of leukemia cells (a mix of cell lines and primary cells) correlated with the ratio of the transcription variants v1 and v2 [37]. Due to the absence of the exon 2 in the variant v2, the resulting protein product lacks the immunoglobulin V-like domain, which is responsible for the interaction with PD-1 [44]. The protein isoform b encoded by the variant 2 was reported to be present in the intracellular membranes, but not on the cell surface, when it was overexpressed in the leukemia cell line K562 [45]. The function of this isoform is still unknown, but it has to be at least partly different from that of the full-length isoform a, which is encoded by the variant v1. One can speculate that v2 isoform might regulate surface exposition and/or function of the isoform a. In the present work, we first validated the correlation of v1/v2 ratio from cDNA samples with the flow cytometry results, for a set of 36 samples of patients with AML at diagnosis. The samples were obtained from leukapheretic products and were thus from patients having hyperleukocytosis at the time of AML diagnosis. As it is shown in Figure 2, we confirmed that the samples, that were PD-L1-positive by flow cytometry, were best discriminated by the v1/v2 ratio. We thus used this parameter for subsequent evaluation of PD-L1 prognostic value.

The patient cohort was increased by 50 cryopreserved cDNA samples. For consistency, the samples were selected to include only patients with hyperleukocytosis who were indicated to leukapheresis at diagnosis. Retrospective analysis of the overall survival revealed significant prognostic impact of the PD-L1 v1/v2 ratio in the whole cohort (Figure 3a). As the mutational status of the *FLT3* gene is an important known prognostic factor in AML, we further evaluated PD-L1 significance in patient subcohorts with the wild-type *FLT3* form (FLT3-WT) and with *FLT3* internal tandem duplications (FLT3-ITD), which are the most frequent *FLT3* mutations, leading to constitutive activity of the kinase. As it is shown in Figure 3b,c, the prognostic value of PD-L1 was limited to FLT3-ITD group. PD-L1 surface expression can be induced both by cell-extrinsic mechanisms, such as proinflammatory cytokines [46,47,48] or hypoxia [49,50], and by cell-intrinsic activation of some signaling pathways [51,52]. For example, a high PD-L1 surface positivity is associated with the expression of some fusion proteins, like NPM-ALK [53] or EML4-ALK [54]. The *FLT3* gene encodes a tyrosine kinase receptor, which plays a key role in controlling survival, proliferation, and differentiation of hematopoietic cells. The internal tandem duplication in FLT3 results in an overactivation of several signaling pathways, including JAK/STAT5 and PI3K/Akt [55,56]. Importantly, oncogenic mutation of JAK2 in myeloproliferative neoplasms was recently shown to induce surface PD-L1 expression through increased promotor activity, and this transformation of myeloid cells was associated with a reduction of T-cell metabolic activity and proliferation [57]. Also, murine leukemia stem cells transduced with FLT3-ITD-coding gene had increased amount of surface PD-L1 after in vitro differentiation into dendritic cells [58]. In our study, we have noted a higher percentage of PD-L1-positive cases in the FLT3-ITD subcohort, although the difference was not statistically significant at the 5% level (*p* = 0.073). It is thus possible that the constitutive activity of FLT3-ITD promotes PD-L1 surface expression, which is likely maintained also on the cells of the residual disease. On the other hand, patients with FLT3-WT might have PD-L1 expression stimulated by cell-extrinsic mechanisms, which are eliminated in the remission state. This would allow for eradication of the residual disease by the immune system, and PD-L1 positivity at diagnosis would not affect the overall survival. 

PD-L1 is known to inhibit T-cell function by binding to its PD-1 receptor. However, the role of PD-L1 in cancer progression may be more complex and involves also the delivery of intrinsic prosurvival signals and protection against the apoptosis [52,59]. The prognostic impact of PD-L1 in FLT3-ITD AML could thus also be due to an increased resistance of leukemia cells to chemotherapy. Nevertheless, our results indicate that PD-L1 may have a prognostic impact on survival specifically in patients with concomitant *NPM1* mutation (Figure 4). An isolated *NPM1* mutation has positive prognostic effect, the underlying mechanism being still unclear. Our previous work as well as recent results of others suggest that *NPM1* mutation is immunogenic, and better prognosis could be associated with an immune response against nucleophosmin or against its interaction partners [60,61,62,63,64,65]. In this case, the observed negative effect of PD-L1 would probably be due to the inhibition of antigen-specific T-cells rather than to an increased cell resistance to chemotherapy.

Recent works also showed that the function of PD-L1 on the surface of cancer cells can be counteracted by increased expression of PD-1, which binds in *cis* to PD-L1 and prevents its interaction in *trans* with PD-1 on T-cells [66,67]. Although we actually detected PD-1 expression on leukemia blasts in some AML samples, it was less frequent compared to PD-L1, and there was no correlation between PD-L1 and PD-1 positivity (Figure 5a). Similarly, no association was found between PD-1 expression level on T-cells and PD-L1 expression level on the autologous leukemia cells (Figure 5b). In leukemia patients with high peripheral blast count, the peripheral T-cells can be considered as tumor-infiltrating lymphocytes (TILs), as the tumor is present in the peripheral blood. TILs usually present marks of exhaustion, including increased PD-1 surface levels. However, the inhibitory effect of PD-L1 only requires PD-1 to be expressed on the T-cells specifically recognizing leukemia antigens, and it is thus not surprising that possibly increased PD-1 expression in this T-cell subpopulation does not largely change the mean values from the whole T-cell population.

One of the few works focusing on PD-L1 in AML reported increased incidence of PD-L1 positivity in patients with *NPM1* mutation, in particular in the CD34+ cell fraction [16]. However, the patient cohort was rather small in that study (15 NPMwt and 15 NPMmut patients) and we did not detect any significant difference between NPMmut and NPMwt patient groups as to the percentage of PD-L1-positive cases in our cohort. We have also tested possible association between PD-L1 and CD34 positivity, the latter being one of the stemness markers of leukemia cells. In agreement with many previous studies, we found lower CD34 expression on leukemia cells from patients with NPMmut (Figure 6). However, the expression of PD-L1 appears to be independent of CD34 positivity.

## 4. Materials and Methods

### 4.1. Material

Mononuclear cells from the peripheral blood of 36 patients with AML indicated at leukapheresis due to hyperleukocytosis at diagnosis were obtained by separation of leukapheretic products on Histopaque 1077 (Sigma, Prague, Czech Republic). The cells were resuspended in RPMI 1640 medium with 10% fetal calf serum and with antibiotics (100 U/mL penicillin, 100 µg/mL streptomycin), and aliquots were used for RNA isolation and for analysis of surface markers by flow cytometry. Additional analyses of PD-L1 transcripts were performed from cDNA samples, which had been taken for routine clinical analyses and cryopreserved. A written informed consent with the use of biological material for research purposes was obtained from all patients. The study was approved by the Ethics Committee of the Institute of Hematology and Blood Transfusion of the Czech Republic as a part of the research project 16-30268A (June 2015).

Antibodies used were the following: CD45-V450 (#560367), CD4-BUV395 (#564724), CD8-BUV395 (#563795), CD19-BUV737 (#564303), CD123-BUV395 (#564195), CD371(CLL-1)-BB515 (#565926), CD135(FLT3)-AlexaFluor647 (#563494), and CD34-BV786 (#743534) were from BD Biosciences (Prague, Czech Republic); CD38-PE (#1P-366-T100) from Exbio (Prague, Czech Republic); PD-1-APC (#17-2799-42) and PD-L1-PE-Cy7 (#25-5983-42) from Affymetrix, Inc. (San Diego, CA, USA).

### 4.2. Flow Cytometry

Cell aliquots (1 × 10^6^ cells per tube) were washed in phosphate buffered saline (PBS) and resuspended in 50 µL PBS. Antibodies were added (2 µL of each) and the samples were incubated for 30 min at 5 °C. Thereafter, they were washed once in PBS and analyzed on a BD Fortessa flow cytometer.

CD45/SSC dotplots were used to eliminate debris and to gate blasts, lymphocytes and monocytes (if present). The lymphocytes were further gated to T-cells (CD4 or CD8-positive) and B-cells (CD19-positive). The gating strategy is illustrated in Figure 1. The background fluorescence of each cell population was obtained from tubes labeled with markers used for gating only (control tube: CD45, CD4, CD8, and CD19). The test tubes contained also PD-1 or PD-L1 antibody and the background fluorescence was subtracted from the mean fluorescence intensity in the corresponding channels. AML blast markers (CD123, CLL-1, FLT3, CD34, and CD38) were analyzed in a separate tube. At least 30,000 cells were recorded for each tube. The sample was then re-run and gated to acquire a sufficient number of lymphocytes in a separate file.

### 4.3. Real-Time Polymerase Chain Reaction

RNA from 2×10^7^ cells was isolated using the RNeasy Mini Kit (Qiagen, Venlo, Netherlands) and cDNA was generated by the reverse transcription on CFX96 real-time system (BioRad) using SensiFAST cDNA Synthesis Kit (Bioline, London, UK). Template RNA and resulting cDNA quality and concentration were assessed on an ND-1000 Nanodrop system (Thermo Fisher Scientific, Waltham, MA, USA). The relative amount of PD-L1 transcript variants was measured by real-time PCR using SensiFAST SYBR N-ROX Kit (Bioline) and calculated using Bio-Rad CFX Manager Software. Primers were designed to target the exon 2/3 boundary (v1), or exon 1/3 boundary (v2) of PD-L1 transcript variant 1 (NM_014143.3), as we described previously [37]. The sequences of the primer pairs were the following: variant 1 forward ATGGTGGTGCCGACTACAAG, variant 1 reverse GGAATTGGTGGTGGTGGTCT, variant 2 forward TTGCTGAACGCCCCATACAA, variant 2 reverse TCCAGATGACTTCGGCCTTG. For the relative quantification by 2^−ΔΔCt^ method, GAPDH expression was measured as a reference, using GAAACTGTGGCGTGATGGC and CCGTTCAGCTCAGGGATGAC as the forward and reverse primers, respectively.

### 4.4. Statistical Analyses

The Cutoff Finder online tool [38] was used to determine the optimal cutoff values for cDNA v1/v2 PD-L1. Survival curves were generated and evaluated using GraphPad Prism software version 7, the Mantel–Cox *t*-test being used to compute p-values for curve comparison. T-test for contingency tables was performed using Epi Info (Centers for Disease Control and Prevention, USA). 

## 5. Conclusions

PD-L1 expression level, which was determined by PCR as the ratio of transcription variants v1/v2, is an important negative prognostic factor for the acute myeloid leukemia, specifically for patients with FLT3-ITD and concomitant NPM1 mutation. We hypothesize that the constitutive activity of FLT3-ITD induces, in a cell-intrinsic manner, expression of PD-L1, which inhibits antigen-specific T-cell response against nucleophosmin or against its interaction partners. In general, the prognostic significance of PD-L1 expression in AML could depend on the mechanism (cell-intrinsic or extrinsic) of its upregulation.

## Figures and Tables

**Figure 1 ijms-20-02823-f001:**
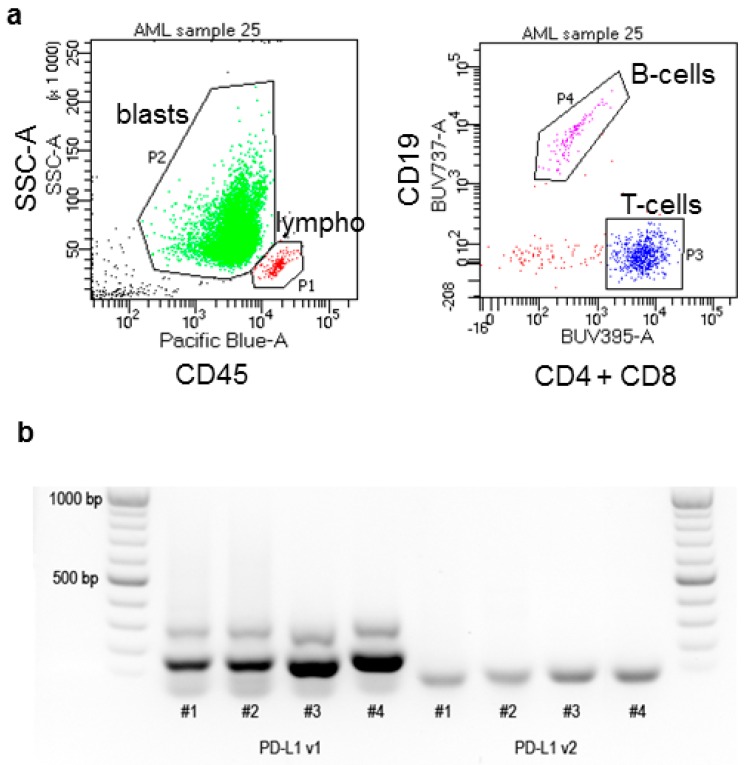
Illustration of the methods. (**a**) Examples of flow cytometry dotplots and the gating strategy. Blasts and lymphocytes were gated in CD45/SSC dotplots (left). B-cells and T-cells were distinguished within the lymphocyte gate P1 (right). (**b**) Example of PCR products obtained using primers for v1 or v2 detection and resolved by gel electrophoresis.

**Figure 2 ijms-20-02823-f002:**
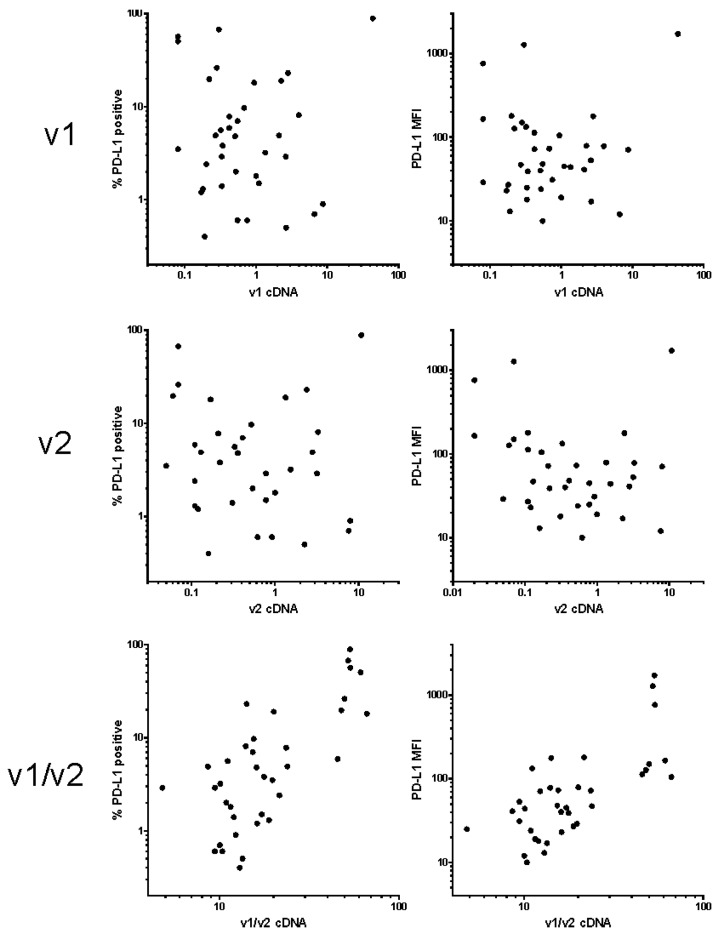
Correlation of PD-L1 cell surface amount with the transcript levels. The amount of PD-L1 on acute myeloid leukemia (AML) blast surface was determined using flow cytometry and expressed as the percentage of positive cells (left column) or as the mean fluorescence intensity (MFI) of the blast population (right column). The relative levels of v1 and v2 transcripts were measured by PCR and normalized to GAPDH.

**Figure 3 ijms-20-02823-f003:**
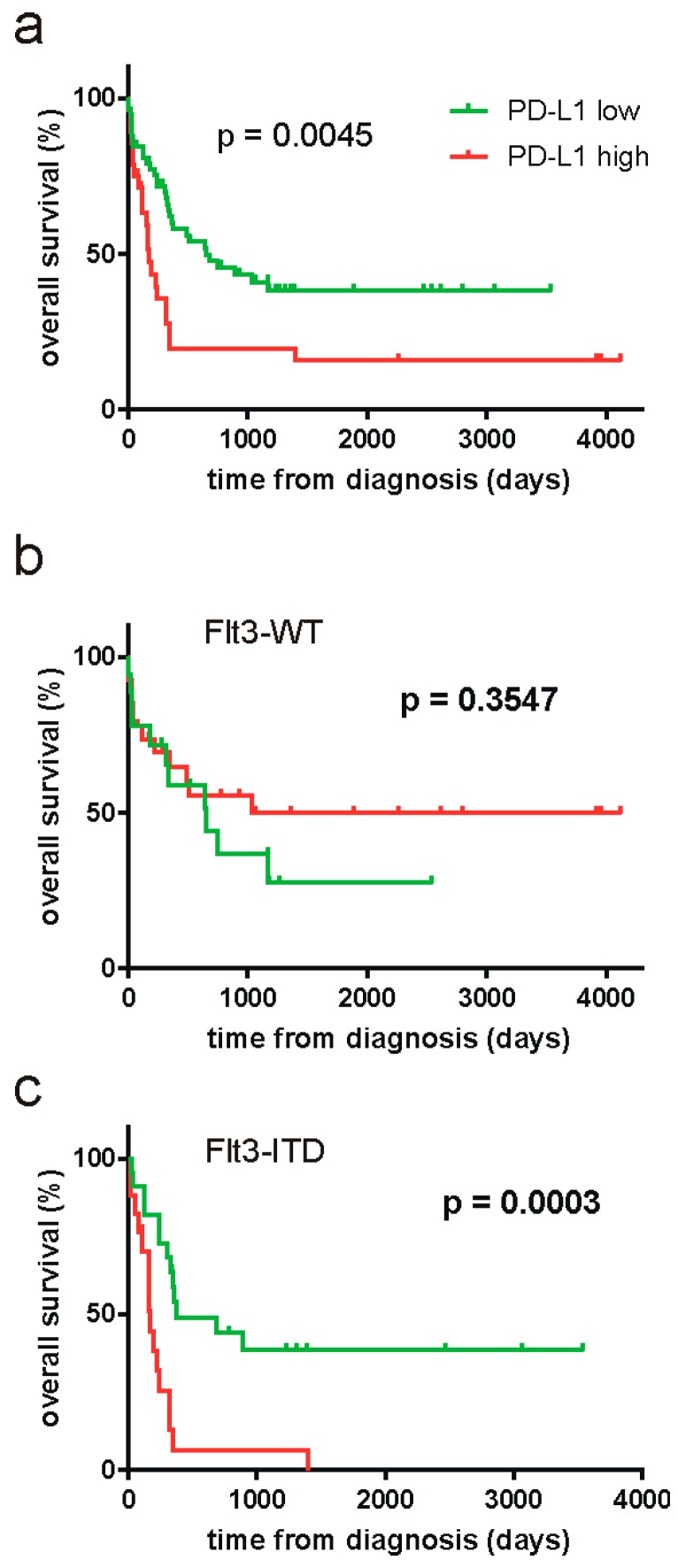
Survival curves for AML patient groups according to PD-L1 levels. The whole cohort (**a**) or the subcohorts containing patients with wild-type FLT3 (**b**) or with FLT3-ITD (**c**) were divided according to v1/v2 PD-L1 cDNA and the corresponding survival curves were compared using the Mantel–Cox *t*-test. The reached p-values are given in the plots.

**Figure 4 ijms-20-02823-f004:**
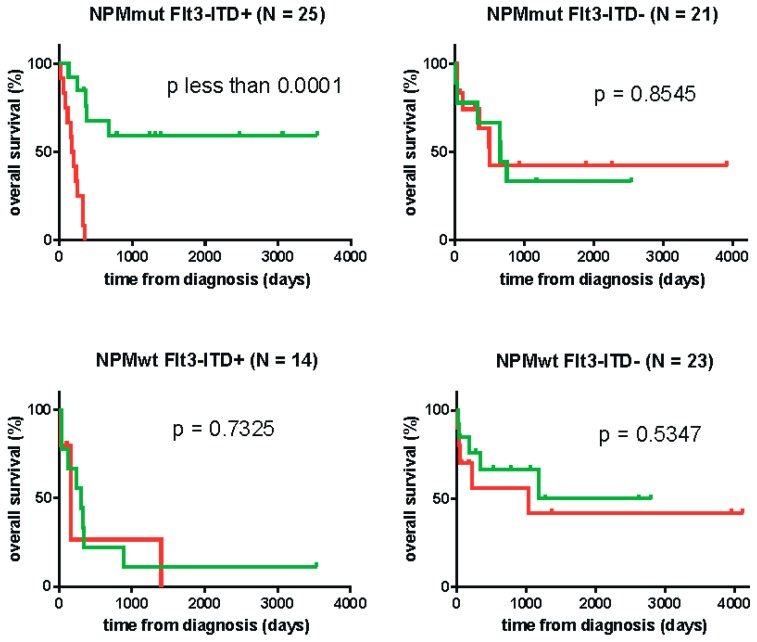
Survival curves for AML patient groups according to PD-L1 levels. AML patients were divided in four subcohorts with different combinations of *FLT3* and *NPM1* mutational status. Each subcohort was subdivided according to v1/v2 PD-L1 cDNA and the corresponding survival curves were compared using the Mantel–Cox *t*-test. The reached p-values are given in the plots.

**Figure 5 ijms-20-02823-f005:**
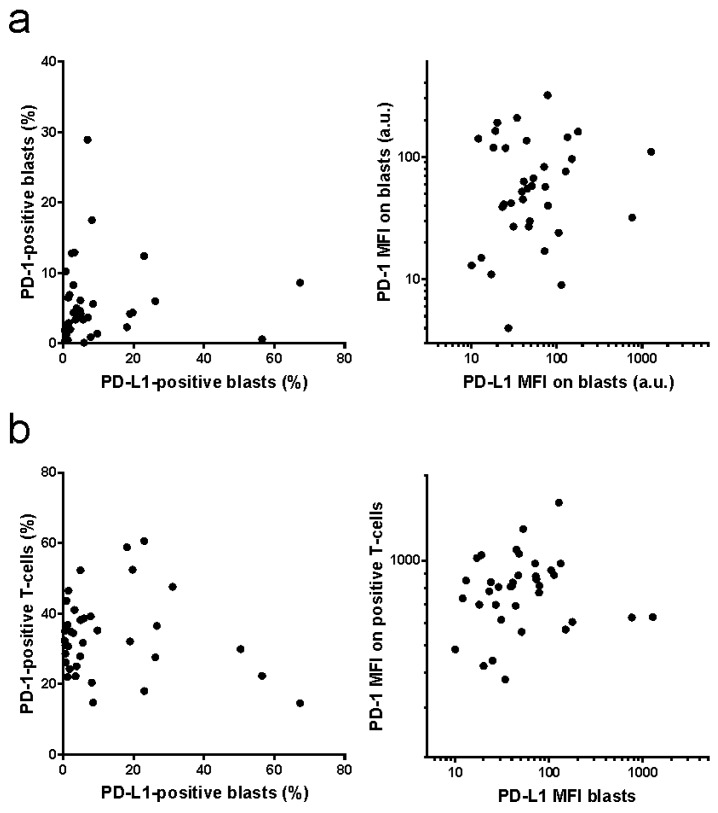
Correlation of PD-L1 and PD-1 expression. PD-L1 expression on AML blasts and PD-1 expression on AML blasts (**a**) or on the autologous T-cells (**b**) was determined by flow cytometry. Left: positive cell fractions, right: mean fluorescence intensity (MFI).

**Figure 6 ijms-20-02823-f006:**
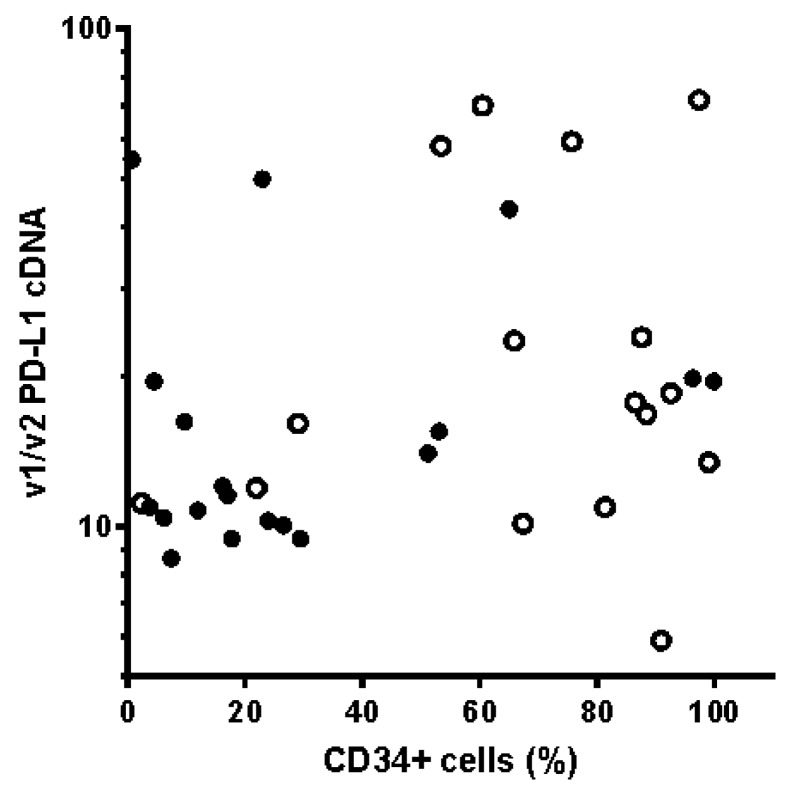
PD-L1 positivity versus CD34-positive cell fraction in leukemia blasts. PD-L1 v1/v2 cDNA levels as a function of CD34-positive cell fraction in samples with the wild-type (empty symbols) or mutated (closed symbols) *NPM1*.

**Table 1 ijms-20-02823-t001:** Review of published studies reporting PD-L1 transcript levels.

Reference	NA Type	Cohort Size	Cancer Type	Primer Targets	Protein Analysis, Correlation with NA	Observed Associations with High PD-L1 Levels
Sasaki 2013 [23]	mRNA	123	NSCLC	v1&v2	not performed	- no difference between tumor and adjacent healthy tissue - higher in pathological T4 types compared to T1
Shen 2014 [24]	mRNA	38	osteo sarcoma	v1&v2	not performed	- correlated with TIL density (*p* = 0.01) - non-significantly worse OS
Yang 2014 [11]	mRNA	124	MDS, AML	v1&v2	IHC in selected 4 patients - correlation (not significant)	- higher expression in MDS, compared to AML - enhancement after hypomethylating therapy was higher in non-responders (*p* = 0.122)
Hassan 2015 [25]	mRNA	26	tuber culosis	v1&v2	not performed	- high in active disease, decrease after therapy
Ikeda 2016 [26]	DNA	94	NSCLC	v1&v2	IHC, FC, no correlation	- non-significant correlation with worse prognosis
Vassilakopoulou 2016 [27]	mRNA	260	SCC	v1&v2	IHC, significant correlation (*p* < 0.001)	- no correlation for mRNA expression - higher TIL density and PD-L1 protein expression correlated with better OS (*p* = 0.059)
Koirala 2016 [28]	cDNA	21	osteo sarcoma	v1&v2	IHC, no correlation	- no correlation for cDNA expression - protein expression correlation with poorer survival (*p* = 0.015)
Kösemehmetoğlu 2017 [29]	cDNA	222	sarcoma	v1 (&v4)	IHC, no correlation	- no correlation of cDNA expression - tendency of higher grade sarcomas to more frequent protein expression
Brüggemann 2017 [22]	mRNA	78	melanoma	v1	IHC, correlation depended on antibody used	- no correlation with response to ipilimumab
Weber 2017 [30]	mRNA	45	OSCC	v1&v2; v1	not performed	- association with malignancy (patients vs healthy donors) - higher expression in peripheral blood of patients with metastasis
Gasser 2017 [31]	mRNA	116	colo rectal CA	not specif.	IHC, significant correlation (*p* = 0.005)	- correlates with advanced stage (*p* = 0.02) and poor prognosis (*p* = 0.004)
Isobe 2018 [32]	mRNA	33	lung adeno CA	v1&v2	IHC, significant correlation (*p* = 0.015)	- no correlation with OS - correlates with shorter PFS after gefitinib therapy (*p* = 0.032)
Amatatsu 2018 [33]	mRNA	124	gastric CA	v1&v2	IHC, correlation with NA not analyzed	- correlation with worse disease type (*p* = 0.024) and lower 5-year survival (*p* < 0.0001)
Tsimafeyeu 2018 [34]	mRNA	473	NSCLC	not specif.	IHC, no correlation with any of the three antibodies tested	
Pawelczyk 2019 [35]	mRNA	62	NSCLC	not specif.	IHC (values 0-1-2), significant correlation (*p* < 0.0001)	- correlation with increased tumor proliferation (*p* < 0.0001), aggressiveness (p = 0.04) and shorter OS in adenocarcinoma group (*p* = 0.033)
Yang 2019 [36]	cDNA	56	T-ALL	v1	IHC, no correlation	- correlated with better OS (*p* = 0.007) - high protein expression correlated with worse OS (*p* = 0.027)

Abbreviations used: NA—nucleic acid, IHC—immunohistochemistry, FC—flow cytometry, OS—overall survival, PFS—progression-free survival, NSCLC—non-small cell lung cancer, MDS—myelodysplastic syndrome, AML—acute myeloid leukemia, SCC—squamous cell carcinoma, OSCC—oral squamous cell carcinoma, CA—carcinoma, T-ALL—T-cell acute lymphocytic leukemia, TIL—tumor-infiltrating lymphocytes, v1/v2/v4—PD-L1 transcription variants.

**Table 2 ijms-20-02823-t002:** Cutoff values and p-values for survival curves in different subcohorts.

Patient Cohort	Cutoff	Reached *p*-Value
all (*N* = 86)	22.72	0.0045
FLT3-WT (*N* = 44)	13.75	0.3547
FLT3-ITD (*N* = 39)	22.72	0.0003
NPMmut FLT3-ITD (*N* = 25)	24.20	< 0.0001
NPMwt FLT3-WT (*N* = 23)	18.84	0.5347
NPMmut FLT3-WT (*N* = 21)	15.07	0.8545
NPMwt FLT3-ITD (*N* = 14)	n.a.	0.7325

The cutoff values for v1/v2 PD-L1 cDNA were obtained from the Cutoff Finder online tool [38]. Patients were divided into PD-L1-low and PD-L1-high groups according to the given PD-L1 cutoff and the survival curves were generated using the GraphPad Prism software. The Mantel–Cox *t*-test was used to evaluate differences between the two groups for each subcohort as indicated.

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
