# Peer review of "High PD-L1 Expression Predicts for Worse Outcome of Leukemia Patients with Concomitant NPM1 and FLT3 Mutations"

_ijms, 2019, doi:10.3390/ijms20112823_

Round 1
Reviewer 1 Report
To the authors:
Despite the work presented here has some potential interest for the readers of the journal, I believe that it would require extensive editing and addressing some scientific issues before publication. Specifically:
1) the article, as it is now, is difficult to follow, contains typos and in many cases the English is incorrect or unscientific (e.g.: line 34 "eficacy"; line 66 "more or less complete remission". What is the meaning of this? There are clinical parameter that define complete remission, either the remission is complete or is not).
2) line 84: positivity cutoffs in some cases are 50% (e.g. Reck et al, NEJM 2016).
3) it is unclear to me if the transcriptional analysis of PD-L1 expression was performed on sorted or on non-sorted blasts. In the second of the cases the RNA would be contaminated by the one of non-malignant cells that could compromise the interpretation of the results. The authors should clarify this point. Also, the author should provide some indication on how homogeneous is the expression of PD-L1 on different cells from the same patient (i.e. are the leukemic cells all with similar MFI?)
4) the authors state several times, even in the abstract, that their working hypothesis is that mutated FLT3 kinase regulate PD-L1. there is no evidence for this in the whole paper, and I don't think this is an acceptable claim without any experiment in which the expression or the activity of FLT3 kinase is manipulate. I believe the paper contains some interesting observations, but I believe that this speculation should be removed as it is completely unsupported. Alternatively, the authors should do some experiment to provide some evidence of this claim.
Author Response
We thank the reviewers for their work and comments. The manuscript was checked for typos and other errors and modified according to reviewer´s suggestions. Specifically:
1) a number of typos were corrected, "more or less complete remission" was replaced by "complete or partial remission" (l.63).
2) The information about cutoff values used for PD-L1 assessment by IHC was precised (Introduction, l.80-82):
"... the positivity cutoff varies from 1 to 10 % among the majority of published studies [19]. In a few cases, the high PD-L1 level was even defined as at least 50 % of positive cells (e.g. in [20])."
3) We described in more detail the results of flow-cytometry measurement (l.131-141):
"The percentage of blasts in the samples obtained from leukapheretic products was at least 52 % (mean: 90%, median: 92%). The signal from PD-L1 antibody usually had rather large intensity distribution on leukemia blasts from the same patient and we thus used both the percentage of positive cells and the mean fluorescence intensity (MFI) for the expression level quantification. PD-L1 expression on lymphocytes was more homogeneous, the positive cell fraction was usually 10 to 20 % and MFI ranged from 50 to 100 units. ... The transcript level of PD-L1 variants was determined from unseparated samples, which contained both blasts and lymphocytes. However, the contribution of lymphocytes was probably only minor, due to their low amount (about 10%) and smaller size, compared to the blasts."
4) We removed the working hypothesis from the abstract and from the introduction. On the other hand, we believe that it is appropriate to keep the idea in the other parts of the MS as it provides a plausible explanation of our findings and should stimulate further research in this field. The hypothesis is clearly phrased as such and is indirectly supported by results of other groups (cited in the discussion).
Reviewer 2 Report
This is an interesting and well-written MS. Overexpression of PD-L1 on tumor cells is mediated via a number of intracellular signaling which promote tumor growth and development. The role of PD-L1 in hematological malignancies is less explored that is why the research taken up by the Authors is extremely important. The applied methods and statistical analyzes do not raise any objections. I strongly recommended this MS to be published in IJMS.
Author Response
Thank you for your work and kind comment.
Round 2
Reviewer 1 Report
I think my concerns have been properly addressed by the authors. I will support publication of this work now.